# The Current Concept of Paternal Bonding: A Systematic Scoping Review

**DOI:** 10.3390/healthcare10112265

**Published:** 2022-11-11

**Authors:** Daichi Suzuki, Yukiko Ohashi, Eriko Shinohara, Yuriko Usui, Fukiko Yamada, Noyuri Yamaji, Kiriko Sasayama, Hitomi Suzuki, Romulo Fernandez Nieva, Katharina da Silva Lopes, Junko Miyazawa, Michiko Hase, Minoru Kabashima, Erika Ota

**Affiliations:** 1Department of Nursing, Faculty of Health and Medical Sciences, Kanagawa Institute of Technology, Atsugi 243-0292, Japan; 2Department of Nursing, Faculty of Nursing, Josai International University, Togane 283-8555, Japan; 3Kitamura Institute of Mental Health Tokyo, Shibuya-ku, Tokyo 151-0063, Japan; 4Research Institute of Imperial Gift Foundation Boshi-Aiiku-Kai, Minato-ku, Tokyo 106-8580, Japan; 5Department of Nursing, School of Medicine, Yokohama City University, Yokohama 236-0004, Japan; 6Department of Midwifery and Women’s Health, Division of Health Sciences & Nursing, Graduate School of Medicine, The University of Tokyo, Bunkyo-ku, Tokyo 113-0033, Japan; 7Department of Midwifery, St. Luke’s International University, Chuo-ku, Tokyo 104-0044, Japan; 8Global Health Nursing, Graduate School of Nursing Science, St. Luke’s International University, Chuo-ku, Tokyo 104-0044, Japan; 9Department of Nursing, Global Health Nursing, International University of Health and Welfare, Narita 286-8686, Japan; 10Gender Studies and Criminology Programme, School of Sociology, University of Otago, P.O. Box 56, Dunedin 9054, New Zealand; 11Graduate School of Public Health, St. Luke’s International University, Chuo-ku, Tokyo 104-0045, Japan; 12Pediatric Nursing, Department of Nursing, Faculty of Nursing, Musashino University, Koto-ku, Tokyo 135-8181, Japan; 13Tokyo Foundation for Policy Research, Minato-ku, Tokyo 106-6234, Japan

**Keywords:** paternal bonding, father–infant relationship, bonding development, fathers, scoping review

## Abstract

Bonding is crucial to perinatal mental health. Despite an extensive body of literature on maternal bonding, few studies have focused on paternal bonding. This scoping review aimed to clarify the current state of the concept of paternal–infant/fetus bonding. The eligibility criteria were drawn from the population concept and context elements to answer the following questions: “what is paternal bonding?” and “what are the constructs of the concept of paternal bonding?” The review comprised 39 studies. Paternal bonding was associated with both positive and negative paternal behavior and thought and may be determined based on fathers’ beliefs and rearing history. Most studies showed that father–child interaction is one of the factors promoting paternal bonding. However, fathers generally felt more distant from their babies post-delivery than mothers. Only a few studies originally defined paternal bonding; most relied on the definitions of maternal bonding. We found different descriptions lacking consensus. Few studies examined the differences between paternal and maternal bonding. No consensus exists on the concept, constructs, and assessment of paternal bonding. The causal relationship between paternal bonding and other variables is unexplored. Future studies should explore fathers’ perspectives and experiences, focusing on the unknown aspects of paternal bonding identified in this review.

## 1. Introduction

Bonding is generally recognized as an emotional tie toward the child, which occurs during the process of becoming a mother. Moreover, bonding lasts an entire lifetime, and these connections include an intense maternal–infant tie involving affective and behavioral domains, not just genealogical connections [1]. Bonding is distinguished from the attachment that a child shows to a particular caregiver. Klaus and Kennell [2] defined *bonding* as a phenomenon of correspondence with parents’ interest and affection for their children rather than a mutual affection between parents and children. Bonding and attachment can be viewed as complementary, such that the lack of either interferes with voluntary parent–child interactions [1,3].

Parent–child bonding has been reported to promote children’s cognitive neurodevelopment [2,4] and enables the parent–child bonding to mature into a better parent–child relationship when it is stronger [5]. Additionally, it is an essential component of a child’s development and social well-being. While most parents are aware of “love and care” for their children, there are some parents who may not feel affection for their children, which may sometimes result in hostility or aggression toward the children. Kumar [6] and Brockington [7] reported cases of mothers who rejected or could not give affection to their babies.

In previous studies, parents with bonding problems, such as impaired bonding, showed poor interaction with their children [8,9]. Poor maternal bonding has been associated with child abuse [10,11,12]. Ample studies have explained the negative developmental consequences of child abuse and neglect. Therefore, impaired bonding could be more serious when children are younger. Due to the significance of the bonding process in parenting, the focus of perinatal mental health research has shifted from perinatal depression to perinatal bonding and bonding disorders [13,14]. Most prior studies have focused on maternal bonding, with limited studies on paternal bonding. Although the child’s main caregivers are both the father and mother, the parenting involvement differs between men and women. Moreover, there are biological role differences at play, such as in childbirth and direct breastfeeding. In addition, while providing childcare, different brain regions are activated in fathers and mothers [15]; therefore, findings on maternal bonding may not apply to paternal bonding. Paternal bonding should be examined as distinct from maternal bonding. Nevertheless, available evidence regarding paternal bonding is limited compared with that on maternal bonding. According to a review focusing on the transition of fatherhood [16], fathers whose partners were pregnant or postpartum described feeling excluded and standing on the outside looking after their partners and following the events. Especially during delivery, they felt powerless and tormented because they could not identify their roles. However, after the baby was born, they felt surprised, loved, and responsible for their baby. They also expressed that they felt the changes in themselves and recognized their responsibility as a parent.

Another review focusing on fathers’ experiences during their partners’ pregnancy reported that fathers who had a pregnant partner felt distant, alienated, and excluded by health care professionals due to their inability to directly feel the fetus. Some of them also felt that a support system is needed for fathers before childbirth because attention and support were directed only toward women and the fetus before birth. They also described that they had positive and negative emotions after delivery and wanted information and involvement in their partner’s birth care. They also wished to be respected and treated as a person rather than be excluded or ignored [17]. Based on these findings, fathers may have experiences with their children that differ from those of mothers. Hence, they may have unique emotions toward their children. Thus, we hypothesized that the concepts and constructs of bonding vary between fathers and mothers. However, unfortunately, there is limited information to determine our hypothesis.

Therefore, scoping concepts and constructs of paternal bonding could be useful to provide more effective support to families and not only to fathers. While there are a few conceptual analyses or maternal bonding studies [1,14], to the best of our knowledge, there are no systematic reviews on the concept of paternal bonding. Furthermore, there is only one integrative review regarding the factors and interventions influencing early father–infant bonding [18]. This review revealed the fathers’ adjustment and transition, influencing variables of father–infant bonding, and promotional interventions to encourage father–infant bonding. Additionally, it described the importance of support, especially during childbirth and transitioning to being a father, associated with the father–infant bonding. However, each study included in this review did not consistently define paternal bonding. Moreover, paternal bonding and other related concepts, such as fatherhood, attachment, the father–infant relationship, and paternal parenting competence, are yet to be clarified. Additionally, the difference between maternal and paternal bonding remains unclear; hence, it is imperative to clarify the concept of paternal bonding. Another related issue is the reliability of the assessment methods for the concept. For example, the scales for maternal bonding should not be used for fathers if the concepts of maternal and paternal bonding are different. In this regard, clarification of the concept can lead to reliable assessment methods and advances in the research field.

To further the advancement of research on paternal bonding, it is also necessary to understand current trends and identify existing gaps in the literature on paternal bonding. Therefore, this scoping review aimed to clarify and integrate the existing concept of paternal–infant/fetus bonding and its constructs.

## 2. Materials and Methods

This review used the Preferred Reporting Items for Systematic Reviews and Meta-Analyses extension for scoping reviews (PRISMA ScR) checklist [19]. It was structured into the following five steps in a methodological framework [20]: (1) identifying the research question(s) (RQ); (2) identifying relevant studies; (3) study selection; (4) charting the data; and (5) collecting, summarizing, and reporting the results.


*Step 1: Identifying the research question(s)*


The following research questions were included.

RQ 1. What is paternal bonding?

What is its definition?What are the measurement tools used for paternal bonding?What terms are consistent with the use and meaning of paternal bonding? Do they have the same use and meaning as maternal bonding?Is there clarity regarding the differences between the concept and other related concepts such as maternal bonding or attachment?

RQ 2. What are the constructs of the concept of paternal bonding?

Are these the same as maternal bonding, or are there unique elements or functions in paternal bonding?


*Step 2: Identifying relevant studies*


The searches were conducted comprehensively using PubMed, CINAHL, EMBASE, PsychINFO, and Medline electronic databases in May 2020. There were no search restrictions. The following terms were used: “paternal”, “infant”, “fetus”, “attachment”, and “bonding” (Appendix A). The eligibility criteria to identify relevant studies were drawn from the population concept and context (PCC) elements [21] of the inclusion criteria below. The manual search was conducted on PubMed and Google Scholar by checking the reference lists of included studies in January 2021.

Population

This study included men who were already fathers or expecting to be fathers. Non-human studies and studies that did not describe fathers or infants were excluded. Studies that described the relationships between parents and infants were included if they represented fathers.

Concept

This scoping review examined the concepts and constructs of paternal bonding. Paternal bonding is an unclear concept and there is no consensus on its pre-existing definitions. For this review, we set our working definition of paternal bonding as “a tie or bond that normally develops between fathers and their fetus or infant.” Bonding has been described from various perspectives: emotional/affective, cognitive, behavioral, and attitudinal. The outcomes were as follows:(1)The current concepts of paternal bonding (the definitions and various assessment methods) and(2)The constructs of paternal bonding.

Context

Any relevant study, no matter the method (qualitative or quantitative), was included. Studies were required to extract conceptually driven definitions/descriptions rather than those needed for statistical analysis. However, studies were excluded if the word “bonding” was not used. Original studies, commentaries, and letters published by a journal were included, while books and theses/dissertations were excluded.


*Step 3: Study Selection*


First, several dyads were assigned independently for the first screening process. Next, full-text screening and data extraction for the initial dyads plus five additional dyads were independently conducted. When discrepancies occurred during these processes, the authors discussed them and resolved the problem. This information was reported based on the PRISMA flow diagram [22].


*Step 4: Charting the data*


The research team conducted the data charting process to come up with a list that summarized the following details: author(s), year of publication, study location, study population, study design, data collection method, outcome measurement tools, results, and key findings. The data tables were created independently by each author.


*Step 5: Collecting, summarizing, and reporting the results*


We summarized the descriptions and created categories based on our study questions and the included studies. We reported the results in a narrative following the created categories and integrated prior knowledge regarding the concept and constructs.

## 3. Results

### 3.1. Description of the Studies

The results of the screening process are shown in a PRISMA flow diagram (Figure 1). First, a total of 8316 records were identified through the database search, and 4560 duplicates were removed. In addition, 21 records were identified via manual searching. Next, 3608 records were excluded because they were irrelevant to the research question. Finally, the remaining 169 studies were selected for full-text screening, and 39 studies were selected based on the eligibility criteria. Overall, 130 studies were excluded, described as follows: 34 records were from an ineligible publication type, 24 records had an incorrect population, 52 records did not match the concept, and 20 records did not match the context. All the included studies are listed in Table 1. Based on all the included studies, we created three themes and five categories (shown below) and reported the results in a narrative.

### 3.2. Theme 1. What Is Paternal Bonding?

#### 3.2.1. Definition of Paternal Bonding

Although a few studies originally defined paternal–infant bonding [23,24,25,26,27,28,29], most studies used a definition that was not distinguished from maternal bonding. All descriptions are shown in Table 2.

Several papers used bonding and attachment interchangeably. Moreover, linguistic confusion existed with no consensus [30]. Some authors provided different definitions of bonding and attachment and claimed that bonding is a parental affective tie toward the child [31,32]. Taylor et al. [33] commented that bonding was used to describe how a mother felt toward her baby, while attachment included the baby’s behavior toward the mother. Pretorius et al. [34] noted that the two terms imply different parental feelings toward the fetus and infant, thus noting that “bonding should be used to refer to the parent’s feelings toward the fetus before birth and that attachment be reserved for the postnatal period.” The most common definition of bonding was a description regarding affective or emotional ties [24,27,28,30,35,36,37,38,39]. The parental belief about the importance of a parental bond with one’s child is what some authors emphasized in the definition of bonding [26]. Some authors extended the definition of bonding to parental behavior [23,24]. They underlined the specific repertoire of parental emotions and behaviors toward the child. Therefore, parental bonding has been defined from emotional/affective, cognitive, and behavioral aspects.

Some authors also disagreed regarding the beginning of parental bonding. They claimed that bonding commences shortly after childbirth [40], whereas others opined that it starts during the fetal stage [24,41]. Although parental bonding may be viewed as having parent-specific emotional, cognitive, and behavioral characteristics, it may also be a mutual process between the parent and child [42]. In this context, both parent and baby should adapt to a new changing situation.

Some studies defined paternal–infant bonding as the unique process of fathers’ attachment to their babies and their relationship [29]. Kerstis et al. [32] described that “bonding reflects a process originating from the parent directed toward the infant [43]; this should not be confused with attachment, which is reciprocal with the infant’s proximity seeking [44].” It was also observed that this process developed slowly through interactions and was established during the perinatal period with biological, psychological, and social factors involved [24,25].

#### 3.2.2. Measurement Tools

A total of 14 tools were used to assess paternal bonding. Of these, two major tools are the Paternal Antenatal Attachment Scale (PAAS [45]) and the Postpartum Bonding Questionnaire (PBQ [46]). The PAAS [27,28,35,36,41,47] and the PBQ were used [32,37,38,48,49,50]. Hoffenkamp et al. [38,49] assessed paternal bonding by combining the My Baby and I questionnaire (MBI [51]), the questionnaire version of the Yale Inventory of Paternal Thoughts and Actions (YIPTA [52]), and the Pictorial Representation of Attachment Measure (PRAM [53]). Other tools included the Bonding Scale [24,42], the Portuguese version of the Mother–Infant Bonding Scale (MIBS [33]), the Taiji Kanjyo Hyotei Syakudo (TKHS [25]) [54]), the Japanese version of the Mother–Infant Bonding Scale (MIBS-J [55]) [39], the validated Maternal-Fetal Attachment questionnaire (MFA [56,57]) [34], the Father-to-Infant Bonding Scale [42,58], the Paternal Fetal Attachment Scale [57]; [59], the Role of the Father Questionnaire (ROFQ [60]), and the Father–Infant Bonding Beliefs Questionnaire (FIBBQ [61]) [40]. The “parental attachment to the child” subscale of the Parenting Stress Index (PSI [62]) was also used to measure bonding [63].

Other studies assessed paternal bonding qualitatively instead of using a quantitative measurement tool. Of these, ten studies used interviews [23,26,63,64,65,66,67,68,69,70], two used group discussion methods [71,72], and three observed fathers’ interactions or reactions [29,31,73]. A study based in Japan measured paternal bonding via a mixed-methods approach. It combined a self-report questionnaire and observed fathers’ behavior through video records and journals [25]. A study based in Belgium also combined video recordings at delivery and interviews [74]. Another study used a smartphone app called HAPPY, which means Handy Application to Promote Preterm infant happY-life [75] and a questionnaire [76].

### 3.3. Theme 2. The Components Related with the Concept of Paternal Bonding

#### 3.3.1. Occurrences of Paternal Bonding

These studies reported some factors that contributed to the occurrences of paternal bonding. Father–infant bonding was determined based on beliefs regarding the importance of the father–infant bond [26], which related to negative paternal bonding with parents’ perception of their child-rearing history [37] and fathers’ experience of neglect [41]. Additionally, fathers’ traumatic experiences during the perinatal period may have contributed to bonding problems [72].

Raphael-Leff [71] divided fathers into “participators” and “renouncers” of pregnancy and parenthood. For participator fathers, the ultrasound examination was essential as fetal movement presented a physical reality. This also helped in father–infant bonding before birth. It was also reported that the gender of a baby, determined by the ultrasound examination, did not affect paternal–infant bonding. Some fathers had already begun bonding with their baby before the gender reveal [73].

Some fathers had ambivalent emotions before delivery [25], and a significant number reported negative emotions such as fear of their babies [24]. By contrast, when meeting their babies for the first time, fathers felt pleasure and relief for their baby’s safe birth. Furthermore, they felt encouraged and responsible as fathers. In addition, the joy felt during the first encounter with the newborn was considered an essential event in father–infant bonding [25]. In another study, Brady et al. [23] divided paternal bonding constructs into two types: “physiology-focused” and “time-focused.” Physiology-focused fathers also felt a bond with their babies; however, it was delayed for six to eight months.

**Table 1 healthcare-10-02265-t001:** All inclusion studies in this review.

Author (Year)	Study Location	Study Population	Methodology Study Design	Data Collection	Bonding Measurement Tools (If Used)
**Avery et al. (2011)** **[64]**	USA	40 men	Qualitative study	Semi-structured focus group interview	n/a
**Brady et al. (2017)** **[23]**	Australia	100 fathers	Qualitative study	Semi-structured interview	n/a
**Brandão et al. (2012)** **[42]**	Portugal	105 fathers	Quasi-experimental study	Self-report questionnaires	The Bonding scale (Portuguese version of the mother-to-infant bonding scale)
**Cheng et al. (2011)** **[63]**	Canada	24 infant–father dyads	Experimental study	Self-report and semi-structured interview	The Parenting Stress Index (PSI) Parent domain: attachment to the child
**Dayton et al. (2019)** **[41]**	USA	51 fathers	Part of longitudinal study	Self-report questionnaires	Maternal/Paternal Antenatal Attachment Scale (MAAS/PAAS)
**de Cock et al. (2016)** **[35]**	Netherlands	247 fathers	Longitudinal cohort study	Self-report questionnaire	Paternal Antenatal Attachment Scale (PAAS) Paternal Postnatal Attachment Scale (PPAS)
**de Cock et al. (2017)** **[47]**	Netherlands	261 fathers	Prospective longitudinal cohort study	Self-report questionnaires	Maternal/Paternal Antenatal Attachment Scales (MAAS/PAAS)
**de Montigny et al. (2018)** **[26]**	Canada	43 fathers	Qualitative study	Semi-structured interview	n/a
**Edhborg et al. (2005)** **[48]**	Sweden	106 fathers	Cross-sectional study	Self-report questionnaires	Postpartum Bonding Questionnaire (PBQ)
**Figueiredo et al. (2007)** **[24]**	Portugal	315 mothers and 141 fathers	Mixed method study	Interview and self-report	The Bonding scale (Portuguese version of the new mother-to-infant bonding scale)
**Freeman et al. (2000)** **[73]**	USA	25 fathers	Cross-sectional study	(1) Questionnaire (2) Interview and observation	n/a
**Furukawa et al. (2020)** **[25]**	Japan	27 fathers	Mixed method study	Self-report questionnaire, husband’s visitation records, video-mediated communication records, and journal	Taiji Kanjo Hyotei Syakudo (TKHS) Qualitative description
**Gearing et al. (1978)** **[77]**	USA	Men before, during, and after the birth of the first child	Education and counseling combination programs	n/a	n/a
**Göbel et al. (2019)** **[36]**	Germany	93 couples	Cross-sectional study	Self-report questionnaires in the 2nd or 3rd trimesters of pregnancy	Maternal/Paternal Antenatal Attachment Scale (MAAS/PAAS)
**Guillaume et al. (2013)** **[65]**	France	60 parents (30 each)	Qualitative study	Semi-directive interview	n/a
**Habib et al. (2005)** **[27]**	Australia	115 expectant fathers	Part of longitudinal study	Self-report and interview	The Paternal Antenatal Attachment Scale (PAAS)
**Habib et al. (2006)** **[28]**	Australia	115 men	Longitudinal project	Questionnaire	Paternal Antenatal Attachment Scale (PAAS)
**Hall et al. (2015)** **[37]**	Netherlands	198 fathers	Cross-sectional study	Self-report questionnaires	The Postpartum Bonding Questionnaire (PBQ)
**Hoffenkamp et al. (2012)** **[38]**	Netherlands	200 mothers and 193 fathers	Longitudinal study	Self-report and interview	Pictorial Representation of Attachment Measure (PRAM) PRAM Self-Baby Distance (PRAM-SBD) Postpartum Bonding Questionnaire (PBQ)
**Hoffenkamp et al. (2015)** **[49]**	Netherlands	144 fathers	Randomized control trial	Self-report questionnaire	Postpartum Bonding Questionnaire (PBQ) Yale Inventory of Parental Thoughts and Actions (YIPTA) and My Baby and I questionnaire
**Kerstis et al. (2016)** **[32]**	Sweden	727 couples	Population-based cohort study	Self-report questionnaires	The Postpartum Bonding Questionnaire (PBQ)
**Longworth et al. (2011)** **[66]**	UK	11 fathers	Qualitative study	Semi-structured interviews	n/a
**Mäkelä et al. (2018)** **[76]**	Finland	26 parents 22 mothers and 4 fathers	Qualitative study	Smartphone apps and questionnaire	n/a
**Marrs et al. (2014)** **[67]**	UK	8 fathers	Qualitative study	Grounded theory method Interview	n/a
**Modé et al. (2014)** **[68]**	Sweden	8 fathers	Inductive, qualitative, and descriptive study	Semi-structured interviews	n/a
**Nishigori et al. (2019)** **[39]**	Japan	1008 fathers	Cross-sectional study	Self-report questionnaire	Japanese version of the Mother–Infant Bonding Scale (MIBS-J)
**Olsson et al. (2017)** **[69]**	Sweden	20 fathers of preterm infants	Descriptive design	Semi-structured interview	n/a
**Palkovitz (1992)** **[40]**	USA	35 married couples	Qualitative study	Interview and questionnaire	The Role of the Father Questionnaire (ROFQ) Father–Infant Bonding Beliefs Questionnaire (FIBBQ)
**Pretorius et al. (2006)** **[34]**	USA	65 fathers	Observational study	Self-report questionnaires	Validated and modified version of Maternal-fetal Attachment (MFA) Questionnaire
**Raphael-Leff (1985)** **[71]**	UK	20 couples	Qualitative study	Interview, questionnaire, and group discussion	n/a
**Riera-Martin et al. (2018)** **[30]**	Spain	376 fathers	Part of cohort study (instrumental design)	Self-report questionnaires	Paternal Antenatal Attachment Scale (PAAS)
**Rousseau et al. (2019)** **[74]**	Belgium	17 parents for the Interview 31 birth filmed	Descriptive paradigm of ethological observation	Interview and video recording at the delivery	n/a
**Rudolf et al. (2014)** **[50]**	Germany	368 fathers	Cross-sectional study	Self-report questionnaire	The Postpartum Bonding Questionnaire (PBQ)
**Soares et al. (2019)** **[70]**	Brazil	38 health professions	Qualitative study	2 parts of semi-structured interview with health professions	n/a
**Suchy et al. (2020)** **[58]**	USA	98 fathers	Randomized crossover trial	Self-report and interview after discharge	Father-to-Infant Bonding Scale (Modified version of 8-item Mother-to-Infant Bonding Scale)
**Taubenheim (1981)** **[29]**	USA	14 fathers	Quantitative study	Self-report and observational tool	n/a
**Thomson-Salo et al. (2017)** **[72]**	Australia	n/a	Qualitative study	Group session focus group discussion	n/a
**Toney (1983)** **[31]**	USA	37 fathers	Randomized control trial	Observation for the interaction assessment	n/a
**Wu et al. (1988)** **[59]**	USA	57 couples	Quantitative study	Self-report measures	Paternal Fetal Attachment Scale

Several studies reported father–child interactions and fathers’ direct experience with their babies as the occurrence of bonding. Gearing [77] explained that the initial contact with the baby influenced the factors related to the fathers’ caretaking behaviors, such as the active hour of interaction with the baby after birth, which was critical and contributed to father–child bonding. In addition, fathers felt bonding through positive responses from the babies [23]. Further, skin-to-skin contact was an essential factor in the development of paternal–infant bonding [69]. Therefore, any interaction between the fathers and infants was deemed important.

**Table 2 healthcare-10-02265-t002:** Definition of paternal bonding.

Developmental Changes	
**Palkovitz 1992** **[40]**	“Bonding,” a term introduced by Klaus and Kennel [78], refers to unique developmental changes that occur shortly after birth—changes that exert a lasting effect on the later parent–infant relationship and child development [79]. Although the use of the term “bonding” in professional literature has more recently come to refer to a longer process of parents’ emotional relating to the infant over time [2,80], the more prevalent popular usage [61] is to restrict the meaning of the term to a set of time-limited and irreversible changes in parent–infant relationship occurring shortly after birth [81,82,83]. In this latter form, the concept of bonding suggests that within the first few hours after birth there are “systematic change” in parental responsiveness to infant behavior, and that parents’ “experiences during these first few hours have lasting effects on subsequent (parental) behavior” [79].
**Process of emotional involvement in perinatal**	
**Figueiredo 2007** **[24]**	Bonding has generally been described as a gradual process of emotional involvement that is progressively established during pregnancy and, more specifically, during childbirth [84].
**Process from the parent and directed towards the infant**	
**Taubenheim 1981** **[29]**	Paternal–infant bonding is a process by which the father forms an attachment to his newborn. An attachment is defined as a relationship between two people that is specific, unique, and endures through time [78].
**Kerstis 2016** **[32]**	Bonding reflects a process originating from the parent and directed toward the infant [43]; this should not be confused with attachment, which is reciprocal to the infant’s proximity-seeking [44].
**Process of mutual adaption**	
**Brandão 2012** **[42]**	Emotional involvement with the baby has been described as a process of mutual adaptation between the parents and the baby, gradually established from pregnancy to the first moments after birth, and affected by the biological, psychological, and social context [24]
**Tie between the parent and the infant**	
**Edhborg 2005** **[48]**	The development of the relationship between mother and infant is the most important psychological process following childbirth and disturbances in this process could potentially pose long-term consequences for the child. ’Bonding’ is one of the many typical words used to describe this process [85]. Bonding refers to the tie between the parent and the infant, a relationship that could be defined as unique between two people and is one that endures through time [86].
**Tie to the child beginning in pregnancy**	
**Dayton 2019** **[41]**	As argued by Walsh [3] and adopted by contemporary attachment researchers [35], the terms ‘parental–fetal relationship’ or ‘bond’ more accurately represent the parent’s tie to the child, beginning in pregnancy.
**Feeling toward the fetus**	
**Pretorius 2006** **[34]**	Bonding should be used to refer to the parent’s feelings toward the fetus before birth and that attachment be reserved for the postnatal period.
**Specific repertoire of emotions and behaviors toward the newborn**	
**Figueiredo 2007** **[24]**	Several authors have observed a specific repertoire of emotions and behaviors toward the newborn in most mothers, as early as immediately after delivery, designated as primary maternal preoccupation [87], bonding [78], or maternal attachment [88].
**Brady 2017** **[23]**	Fathers’ descriptions of their bonds were usually simple, positive responses such as ‘it’s good’.
**Affective/Emotional tie**	
**Toney 1983** **[31]**	The early development of the parent–infant relationship is divided into bonding and attachment. Bonding is the rapid information of an affectional tie, unidirectional from the parent to the infant during the first hours and days after birth and enhanced by physical contact [81].
**Habib 2005** **[27]**	“One quality of the father–infant relationship that has not received much attention is the father’s perception of his emotional or psychological bond to the child. Some have referred to this bond as the father’s attachment to the infant [45]. However, the use of the term “attachment” in this context is contentious for it is not consistent with attachment theory’s central concept of psychological dependence. That is, infants are attached to their parents because they are dependent upon them; parents are not dependent on their infants, and hence not strictly attached to their infants. This paper will refer to authors who have used the term father attachment. In doing so, we apply the notion of father attachment in the sense of the emotional or psychological bond of the father to his infant.
	On the other hand, there may be contexts in which this is not the case that may underscore the difference between instrumental father-involvement and the emotional quality of the father–child relationship as indicated by the father–infant bond. In any case, it will be important to measure instrumental dimensions separately from the father–infant bond and examine relationships between them.”
**Habib 2006** **[28]**	We use the more appropriate concept of the emotional connection or psychological bond of the father to his child, and we refer to this as the paternal–fetal bond.
**Figueiredo 2007** **[24]**	The present study aims to investigate the initial emotional involvement with the newborn, during the first week after delivery, focusing on three main objectives: description of mother-to-infant and father-to-infant initial emotional involvement, differences between mothers and fathers in terms of their initial emotional involvement, and changes in mother-to-infant emotional involvement after the first delivery days.
**Hoffenkamp 2012** **[38]**	The process of bonding, in which parents form an emotional bond or tie with their infant, is thus essential for the infant’s survival and development, as the development of an affectionate parent–infant relationship enhances parental investment [89,90].
**Hall 2015** **[37]**	The quality of the relationship between a parent and his or her infant contributes significantly to the development of the infant [91]. Generally, this relationship has been examined from the infant’s perspective, by examining the quality of infant-to-parent attachment. From the parent’s perspective, ‘bonding’ has been described as the quality of the emotional tie from the parent to the infant.
**de Cock 2016** **[35]**	The parent-to-child bond can be defined as an affective tie from parent to child, which stems from the caregiving system and is aimed at protecting the child [92]. We will use the term bonding to refer to the parent-to-child bond.
**Riera-Martin 2018** **[30]**	Bonding is the term adopted in this study, meaning the unique emotional tie established between parents and their baby in early childhood.
**Göbel 2019** **[36]**	In contrast, Condon [45] defined parental–fetal bonding as focusing on the developing affectionate bond with the fetus and introduced the Maternal/Paternal Antenatal Attachment Scale (MAAS/PAAS) [45].
**Nishigori 2019** **[39]**	Bonding is the parental emotional relationship with their infant(s), and bonding failure has been defined as a parental mental disorder characterized by indifference, reduced affection, anger, rejection, hostile feelings, and/or an impulse to harm their infant(s) [6,93].
**Love**	
**Habib 2005** **[27]**	Condon and his colleagues [45,94] described a phenomenological view of the paternal–fetal (or parental–fetus) bond. According to Condon’s approach, the paternal–fetal bond is a subjective feeling state of love for the unborn child, rather than an attitude or belief about the child, and is at the heart of a man’s (and a woman’s) experience of early parenting. Two dimensions of this bond have been empirically derived: the quality of the bond, which refers to the nature of the emotional experience when thinking about the fetus, and the intensity of preoccupation with the fetus.
**de Cock 2017** **[47]**	Despite the fact that parental bonding is a key component of the caregiving relationship, this concept is still understudied. Parental bonding can be defined as a subjective experience of affection of the parent towards the child. The core of the parental bond is a feeling state (“love”) that eventually exposes itself in parental behavior [45,94]
**Beliefs and reciprocity**	
**de Montigny 2018** **[26]**	The value attributed to early father–infant bonding, characterized by the father’s expressed beliefs about the importance of his bond with his infant and his observations regarding the reciprocity of this bond.
**Multidimensional conceptualization of fathering**	
**Habib 2005** **[27]**	Elucidating an emotional quality of the father–child relationship—the father’s bond to his child—has shown it to be a distinct and potentially important component of fathering.
**Hoffenkamp 2012** **[38]**	The concept of bonding is complex and multi-faceted in origin, yet the PRAM * attempts to provide a visual representation of the relationship between the parent and the baby.
**Furukawa 2020** **[25]**	Fathers’ journal entries focused on three concepts: their new roles, thankfulness, and feelings. The fathers’ first encounter with their newborns was the most significant among all participants.

* *Note.* Pictorial Representation of Attachment Measure (PRAM).

#### 3.3.2. Promotional Factor of Paternal Bonding

Father–infant interactions were reported to be a promotional factor as well as an occurrence of bonding. Previous studies have reported various types of interaction settings and the importance of contact as early as possible during or after delivery. Experiencing umbilical cord cutting positively influenced paternal–infant bonding before and after birth [42]. Fathers had more interaction with babies delivered by cesarean section or forceps than vaginal delivery, and holding their baby earlier after delivery helped increase their bonding behavior, such as interactions with the baby [31]. A Canadian study reported that fathers focused on developing a bond earlier; the interaction time length differed between fathers and mothers, and fathers felt joyful when their babies recognized them [26]. Furukawa et al. [25] reported that communication and early experience/involvement after delivery with their babies were crucial factors for improving and promoting paternal–infant bonding. In addition, taking care of their babies directly, such as by massaging and feeding, and being closer to their babies through affectionate actions such as seeing their face, talking to them more with a monotonous tone, and holding them for long durations were related to the development of a bond [29,58,70,74,76]. For example, bottle feeding was associated with father–infant bonding. It also brought positive emotions for fathers as they realized the importance of bottle feeding [26,64]. Nevertheless, fathers felt distant from their babies; further, they felt that their interaction was unequal as they could not be directly involved in their babies’ breastfeeding [26]. Moreover, another study reported that skin-to-skin contact was a factor that contributed to the development of father–infant bonding. However, fathers were confused during their first encounter with their babies [69]. According to these results, fathers promoted bonding and focused on father–infant connections through skin-to-skin contact [69] and feeding their babies [26]. In addition, fathers wanted the same interactions with their babies as mothers had [26].

Another promotional factor of paternal bonding was family-friendly hospital settings [70]. A U.K. study reported fathers’ negative concerns regarding the distance between them and their babies caused by hospitalization and work [67]. However, precise information regarding what procedures the staff conducted and why, babies’ conditions, expectations for the babies, answering fathers’ questions, and fathers’ involvement in baby care were associated with developing and improving father–infant bonding, especially when long-term care was required [68]. In addition, “the length of initial father–infant contact with the infant,” “mothers’ reported satisfaction with the initial father–infant contact,” “fathers’ satisfaction with their role in labor,” “the total amount of time the father spent with the infant prior to discharge from the hospital,” and “the number of staff present at the delivery” were associated with improvement in father–infant bonding [40].

Other promoting factors included 3D and 4D ultrasound examinations [34]. Real-time and visualized information was also helpful in increasing father–infant bonding compared with picture information. This approach could enhance initial bonding [73]. Another study reported that support from the partner increased father–infant bonding [47].

#### 3.3.3. Associations of Paternal Bonding

Paternal–infant bonding has been associated with paternal behaviors and thought, both positively and negatively. Positive paternal bonding promoted interactions with their babies and contributed to maternal psychological well-being [65]. However, some studies have reported negative effects. For example, while most fathers interacted with their babies, some also reported that watching from a distance was enough, as they were afraid to hurt their babies [65].

Paternal–infant bonding was also associated with psychosocial aspects of the father’s self or the family. Further, negative paternal bonding development caused negative emotions toward their baby. Negative paternal bonding development was associated with maternal and paternal postpartum depression, deteriorated marital relationship experiences, paternal physical and mental health conditions, and negative maternal bonding development [32]. Lower paternal bonding or impaired bonding with their babies was associated with paternal and maternal stress, impaired attachment [50], unstable emotion similar to that experienced by lower-bonding mothers, negative emotion and control toward their babies, perceived additional difficulties in child temperament, and anxiety [35].

A Japanese study reported that fathers’ bonding problems were associated with their mental health history and intimate partner violence toward pregnant mothers [39].

In addition, attachment style, older siblings, and fathers’ educational level and attributes were associated with paternal bonding. There was a negative association between paternal bonding and avoidant attachment style in fathers who already had babies [36,76]. Thus, the first child may be associated with higher bonding in fathers [35]. Fathers’ educational level was also positively and negatively associated with paternal bonding. More father–infant interactions were observed among fathers with higher education [31]. Meanwhile, it was also reported that fathers with higher education had more bonding problems [37]. However, according to another study, the father’s age, baby’s gender, attendance of prenatal class and birth, and baby caregiving knowledge and experience were not associated with bonding behaviors [29]. By contrast, another study described increased interaction and more instances of smiling by fathers of male babies [31].

### 3.4. Theme 3. Differences between Paternal and Maternal Bonding

Some studies described the differences or similarities between paternal and maternal bonding. We found some characteristics of paternal bonding in the included studies.

Of them, two studies used PBQ to measure postpartum bonding. The study from Sweden reported no differences in the paternal and maternal PBQ and subscale scores at one week postpartum [32]. According to Edhborg et al. [48] a few fathers and mothers showed a “mild bonding disorder” [46], and their scores on “impaired bonding” in the subscale of PBQ were above the cut-off point at one week postpartum. In other subscales of the PBQ, “rejection and anger” and “anxiety regarding care” scores were not above the cut-off point for both paternal and maternal bonding. The study conducted in the Netherlands measured both paternal and maternal bonding using the PBQ and explained that both parents’ perceptions of their parenting experiences strongly affected parent–infant bonding [37]. Another study measured parental bonding using MAAS and PAAS three times before and after delivery. There was a weak correlation between paternal and maternal bonding at all the measurement points [47]. A study in the Netherlands measured parental bonding with hospital-based video interactions. The hospital-based video interaction guidance had positive effects on parental to preterm infant bonding, especially for fathers and mothers with a traumatic experience of preterm birth [49].

Paternal and maternal bonding also showed differences. For example, an Australian study that conducted semi-structured interviews revealed that almost all fathers considered their bonding as time-focused disclosure instead of physiology-focused disclosure. The differences were described based on the caring time for parents and their babies [23]. A Canadian study that conducted semi-structured interviews reported that fathers focused on the intensity of the bond with their babies based on envious feelings toward the mother–infant relationship. Therefore, fathers felt distant from their babies compared with mothers during breastfeeding or other interactions and wanted to relate and experience activities with their babies with the same intensity as mothers [26]. Two other studies described the differences between paternal and maternal bonding when using the PBQ. A Swedish study described the communality between paternal and maternal bonding. However, it was also reported that the maternal total PBQ score at two months postpartum was lower than one week postpartum, although there were no differences in the PBQ score of fathers over time [48]. Conversely, a study from the Netherlands reported that the fathers’ PBQ scores were higher than the mothers’ at both one and six months postpartum, and there was a relationship between emotional bonding between the two time points. Furthermore, another study from Portugal focused on the differentiation of chronological changes in fathers’ and mothers’ bonding scores. This study measured fathers’ and mothers’ emotional involvement with the infant via the bonding scale, the Portuguese version of the new MIBS, and reported that the fathers’ bonding was higher than that of the mothers, and no fathers reported absent bonding. It also described that fathers had better bonding 48 h after delivery than mothers [24]. In another study, mothers of preterm infants reported higher feelings of bonding than those of full-term infants, whereas for fathers, no differences in feelings of bonding were found between the gestational age groups. In addition, not only was the quality of paternal bonding predicted by an earlier reported bonding, but a direct effect of care on bonding problems was also observed. Furthermore, for fathers, a higher educational level and age predicted more bonding problems [37].

Paternal bonding has been reported to be similar to maternal bonding in terms of the ways of developing the bond [95], as cited by Figueiredo et al. [24], and lower quality of parent bonding has been associated with parenting stress and child functioning problems [47]. For instance, paternal bonding, especially at 24 months postpartum, was negatively associated with paternal parenting stress [47]. Most fathers focused on sufficient time to develop a bond with their infant, regardless of the infant’s gender. While few fathers focused on the physiological construction of early development of bonding, such as breastfeeding and maternal instinct, they also felt “useless” in their infants’ lives for a couple of early months after delivery [23]. The changes in fatherhood influenced their new roles and strengthened father–infant bonding [70].

Another study in Spain described the similarities between the constructs of paternal and maternal bonding. This study evaluated paternal and maternal bonding via the Spanish version of the MPAS and PPAS, and then developed a single version of these scales, common for both parents. Consequently, both maternal and paternal bonding scales were constructed using three factors: “quality of bonding,” “absence of hostility,” and “pleasure in interaction.”

## 4. Discussion

### 4.1. Summary of Evidence

Our scoping review revealed the existing concept and constructs of paternal bonding described in the extant literature, which we then reviewed and summarized systematically. In the following section, we discuss the scopes according to our RQs, along with the future prospects.

### 4.2. What Is Paternal Bonding?

The most used definitions of paternal bonding were descriptions of paternal emotional ties. This was similar to the case of maternal bonding, which was often explained by emotions [1]. Notably, some studies used other words such as affect [35,36] or love [27,47] instead of emotion. Habib and Lancaster [27] and de Cock et al. [47], citing Condon and his colleagues [45,94], proposed a definition of bonding that is narrowly limited to parental love toward the child. By definition, this idea excludes emotions other than love (i.e., happiness). Bonding disorder is limited to its deficiency (lack of love). In comparison, using the term emotion/affect provides wider definitions of bonding and may include emotions such as anger, fear, and disgust [24,27,28,30,35,36,37,38,39].

Other studies defined bonding as the specific repertoire of paternal emotions and behaviors toward the infant or the multidimensional conceptualization of fathering. These might suggest broader aspects of the concept of emotional bonding.

While most studies described paternal bonding as the paternal emotional tie toward the infant, others defined paternal bonding as an emotional tie between the father and infant [48]. The latter definition may indicate that parental bonding is a mutual process between the parent and child [42] and may be viewed as having parent-specific emotional, cognitive, and behavioral characteristics. Some studies defined paternal–infant bonding as the unique attachment process of fathers to babies and their relationship [29]. This is an essential view as it implies that parental bonding is a response to cues from the baby as a form of attachment. Nonetheless, both parents and infants play an essential role in adapting to new situations because bonding and attachment involve reciprocal interaction. The attachment is fulfilled or rejected by a parent’s appropriate or inappropriate approach, that is, bonding [44].

Furthermore, the literature has shown inconsistent data on the beginning of parental bonding. Some studies claimed that bonding starts shortly after childbirth [40], whereas others reported that it starts during the fetal stage [24,41]. This is important because the latter view leads to studies on parent–fetus relationships. The literature also explained that this process developed slowly through interactions and is established during the perinatal period with biological, psychological, and social factors [24,25].

Thus, parental bonding has been defined differently. Further exploration is required on whether these synonyms were intentionally distinguished in studying paternal bonding. In summary, paternal bonding was deemed to be a psychological dynamic that originated in a father toward his child. However, there was no consensus regarding the definition. This suggests that the core concept is not identified.

### 4.3. What Are the Measurement Tools Used for Paternal Bonding?

Paternal bonding has been studied in psychology, nursing, social policy, perinatal education, and child and family studies, among others. This implies that there is high interest in this concept in practical and clinical settings. However, there is no established method to assess paternal bonding; therefore, various methods such as behavioral observation, diary journals, interviews, and scales have been used as substitutes. Hence, it should be carefully determined whether each method can assess paternal bonding.

A popular assessment scale for paternal bonding was derived from fathers’ interviews regarding their experiences [45]. However, most assessment scales used in current studies notably adopted a conceptual framework based on maternal bonding studies. Some studies have even adopted maternal bonding scales, while others used validated versions of the maternal bonding scales to study paternal bonding. However, the representation of bonding may differ between men and women. There may be qualitative differences between paternal and maternal bonding in their nature; hence, the constructs of their bonding might also differ. Condon et al. [96] identified the constructs of the Paternal Postnatal Attachment Questionnaire as “patience and tolerance,” “pleasure in interaction,” and “affection and pride,” while those for the maternal version were “the pleasure of proximity,” “tolerance,” “the need for gratification and protection,” and “the acquisition of knowledge” [94]. The constructs of the MAAS indicated the qualitative differences in father–mother bonding constructs and their representations. The core essence of bonding might extend further beyond these differences. Recently, studies with a sophisticated design examined the measurement invariance of the measurement tool between paternal and maternal bonding, before analysis [97]. Nonetheless, we should interpret the findings of previous studies using measurement methods that have not been validated with caution.

Limited studies were conducted to explain the qualitative differences or commonalities between maternal and paternal bonding; therefore, we failed to find specific constructs of paternal bonding. Thus, our scoping review suggests the need for future research on the constructs of maternal and paternal bonding. Clarification of the concept and elucidation of its constructs can lead to the reliability of assessment methods and advance the research in this field.

### 4.4. What Are Consistent in the Use and Meaning?

In the current literature, the most common conceptual descriptions of bonding relate to attachment. Both concepts have often been used interchangeably. However, in our scoping review, we found an intentional distinction between these two concepts in the selected literature that used the word “bonding.” Several studies recognized that bonding initially reflected the parent’s feelings directed toward the infant, whereas attachment reflected the infant’s directivity to their caregivers [31,32,33,43,44].

Researchers have explained that attachment is necessary for biological survival and psychosocial development, particularly in human beings [59]. Another study distinguished attachment and bonding in that the parents were not dependent on their infants and were thus not strictly attached to them [27]. In the current literature, attachment is not accepted as a surrogate term for the maternal or paternal bonding concepts [1,14]; thus, such linguistic confusion should be clarified [30,47]. A particular study explained the boundaries between the two concepts—bonding should refer to the parents’ feelings toward the fetus before birth, and attachment should be reserved for the postnatal period [34]. However, this was a peculiar definition.

No piece of literature has clearly described the distinction between paternal bonding and related concepts such as relationships, ties, and fatherhood.

### 4.5. Is There Clarity in the Differentiation of the Concepts from Other Related Concepts?

Our research interest was the difference between paternal bonding and maternal bonding. Unfortunately, it was difficult to capture the specific concept and construct of paternal bonding because most studies used the same definitions for maternal and paternal bonding. This might imply that the concept of paternal bonding is theoretically undistinguished from maternal bonding. This may be attributed to undeveloped discussions; thus, it is unclear whether the actual constructs of maternal and paternal bonding are precisely the same. Therefore, further research is required to clarify this point.

Regarding constructs related to paternal bonding, the current literature revealed a similarity between maternal and paternal bonding concerning the fathers’ desire to connect and experience activities with their babies the same way mothers do [26], and that paternal bonding developed similarly to maternal bonding [95]. There was a weak correlation between maternal and paternal bonding [47], and the behaviors fathers exhibited toward their infants’ engrossment were similar to those displayed during maternal–infant bonding [31]. Conversely, a study reported a greater distance between father and infant than between mother and infant [26] or a delay and weakness in paternal bonding compared with maternal bonding [37,48]. Given that these studies were conducted in different clinical settings using different assessment methods, we could not directly compare the results. Nevertheless, most studies reported that fathers’ bonding was not fixed but developed (or changed) in a similar way as mothers’ bonding and was harder to occur naturally than mothers’ bonding. It seems reasonable to deduce that paternal bonding development is more challenging than maternal bonding, as mothers can physically experience babies, for example, during breastfeeding. Brady et al. [23] reported that fathers felt “useless” in their infant’s life during the early months after delivery. Moreover, they felt distant from their babies and experienced inequality in interaction during instances such as breastfeeding [26]. Another study suggested that fathers lack knowledge about supporting breastfeeding mothers [64]. Hence, clinical perinatal care should include fathers positively.

However, it should be noted that most measurement tools used in these studies were derived from maternal bonding studies; therefore, they might lack specific aspects of paternal bonding, or might not have captured the essence that was common between fathers and mothers. Further, we could hardly find literature identifying the qualitative differences between maternal and paternal bonding. Thus, the findings of previous studies should be carefully interpreted.

### 4.6. What Are the Constructs of the Concepts of Paternal Bonding?

The literature did not explicitly conclude the constructs of father’s bonding. Based on the integration of the current scientific literature, we recognized the constructs of the concepts of paternal bonding as discussed below.

Paternal emotion toward the child is the core of paternal bonding. Moreover, paternal bonding seems to include paternal cognitions such as beliefs or pride regarding fatherhood. Paternal bonding is often reflected in the fathers’ attitude toward their children or parenting. However, it is not the same as parenting behavior. Biological factors are not sufficient to induce paternal bonding; thus, the occurrence and promotion of paternal bonding may be influenced by psychosocial experiences.

### 4.7. Future Prospects for Studies Regarding the Concept

Our scoping review revealed some crucial gaps in studies related to paternal bonding.

First, as shown in Table 1, the study locations were spread worldwide. This implies that paternal bonding is of global interest, but most studies were conducted with a similar cultural background. Although the study locations were quite diverse, previous studies did not explicitly indicate the cultural aspect of paternal bonding. To our knowledge, no global studies published regard paternal bonding across cultures. By and large, studies cited common definitions not considering the cultural context. Nevertheless, we acknowledge the critical role of culture in paternal bonding. Moreover, various measurement tools were used across studies and cultures. Hence, the measurement invariance of the scale is essential, and further research exploring these areas is needed.

Second, although several studies identified the associated variables of paternal bonding, no study identified the causal relationship between paternal bonding and other variables. Thus, confounding factors were insufficiently examined. Moreover, limited studies clarified the mechanisms underlying paternal bonding.

Third, some clinical studies have reported the promotional factors of paternal bonding. However, experimental studies, including randomized controlled trials, have rarely been conducted. Fourth, only limited studies have examined the consequences of paternal bonding. Lastly, despite the increased interest in mothers’ bonding disorders, there are no reports regarding fathers’ bonding disorders as a mental health problem. Additionally, there have been no studies on the clinical intervention for fathers’ bonding disorders. Thus, it is no exaggeration to say that paternal bonding disorders are entirely unknown. Paternal bonding has been much less researched compared to maternal bonding. Needless to say, in recent years, parenting has not only been for mothers. The literature suggests that paternal bonding affects not only the father–infant relationship but also maternal mental health and child development. Paternal bonding is extremely significant in the context of family dynamics. Due to the lack of experience with physical changes, delivery, and breastfeeding, fathers might require more psychosocial support than mothers. Paternal bonding might be related to parental identification as fathers or their development into this role. However, existing paternal bonding studies have been based on maternal bonding research. Only a few studies focused on fathers’ perspectives to identify what paternal bonding is. Paternal bonding is an essential issue for family mental health, and it is just as important as maternal bonding. However, there is a lack of information that allows the understanding of paternal bonding from fathers’ perspectives and experiences. Therefore, this is a vital topic for future research. Studies focusing on men’s perspectives and voices are necessary to define paternal bonding. Some qualitative studies provided fathers’ direct responses and that method might lead to identifying the common concept of paternal bonding.

### 4.8. Limitation

Since the distinction between the term “bonding” and its synonyms have not been discussed, we may not have been able to screen literature that did not use the term “bonding.” In addition, books were not included or analyzed. Additionally, we did not assess the methodological quality of the studies. Therefore, we could not be sure of the quality of the data/information we collected. Nevertheless, our scoping review successfully identified the concepts of paternal bonding in the current literature, the issue of the studies targeting paternal bonding, and future research implications.

## 5. Conclusions

This is the first review to provide the definitions, measurement tools, and constructs of paternal bonding—in addition to identifying the gaps in the paternal bonding literature—using a scoping review method. We found different ways of defining paternal bonding. Only a few studies originally defined paternal bonding, and most studies used definitions based on that of maternal bonding. Various methods were employed to assess paternal bonding, and few studies have examined the differences between maternal and paternal bonding. Our findings suggest a lack of consensus regarding the concept, constructs, and assessment methods of paternal bonding. Moreover, most of the existing paternal bonding studies have used the concept and measurement tools derived from maternal bonding research. Few studies focused on fathers’ voices to identify what paternal bonding is. Several studies reported that the occurrence of paternal bonding is determined according to the fathers’ beliefs about the importance of a parental bond with a child. Fathers’ early experiences with their babies were also reported as a factor that promotes paternal bonding. It has also been reported that compared with mothers, fathers felt more distant from their babies after delivery. Therefore, supporting early paternal–baby interactions might be key in promoting paternal bonding with the offspring. However, the causal relationship between paternal bonding and other variables is unexplored. Moreover, the qualitative difference or similarity between maternal and paternal bonding has not been clearly described.

Despite these unknowns, the increasing research interest in this concept across various fields indicates the significance of paternal bonding. Existing paternal bonding studies were conducted based on maternal bonding research, highlighting the lack of information about paternal bonding. Further studies based on fathers’ perspectives and experiences are needed, specifically those focusing on the unknown aspects of paternal bonding (as identified in this review) to determine what paternal bonding actually is.

## Figures and Tables

**Figure 1 healthcare-10-02265-f001:**
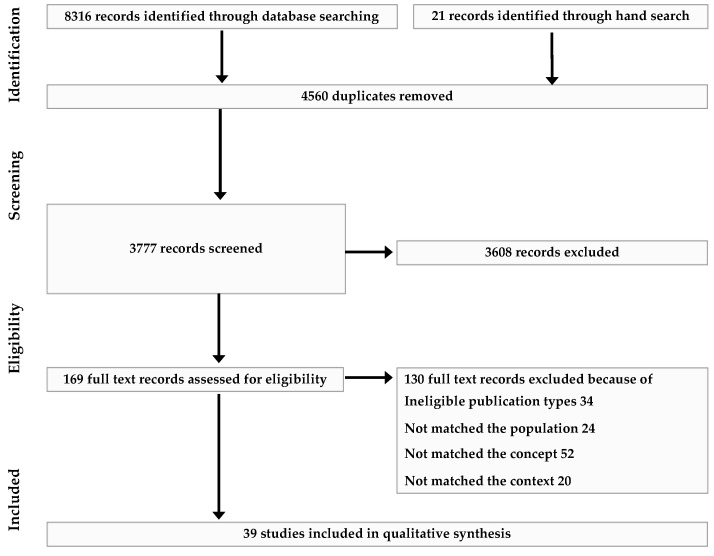
The selection process in PRISMA flow diagram.

## Data Availability

Not applicable.

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
