# Peer review of "The Current Concept of Paternal Bonding: A Systematic Scoping Review"

_healthcare, 2022, doi:10.3390/healthcare10112265_

Round 1

Reviewer 1 Report

Manuscript healthcare-1997280, titled “The current concept of paternal bonding: A systematic scoping review” was submitted to Healthcare for possible publication. This paper is a scoping review of the of the concept of paternal bonding. The authors identified 39 studies and examined multiple research questions regarding the nature of paternal bonding and its correlates.

This manuscript has the potential to provide an important contribution to the literature, as there is not much literature to date on the concept of paternal bonding; much of past work has focused on maternal bonding. Strengths of the manuscript include following a specific review protocol and a comprehensive discussion of the results. My concerns revolve more around the Introduction and rationale for the study. As noted in detail below, a stronger review of past work and evidence for the authors’ claims are needed. Specific comments on sections of the paper are provided below.

Introduction:

-I found the introduction somewhat hard to follow and many of the claims did not feel well-supported. For instance, the authors state “Thus, bonding, a special emotional tie, has different concepts and constructs for fathers and mothers” (line 96) but shortly thereafter state “Additionally, the difference between paternal and maternal bonding remains unclear” (line 106). More evidence needs to be provided to explain how and why bonding differs between mothers and fathers or, if it’s the case that we actually don’t know that, the claim that there are differences should not be made.

-Related, I was unclear how the differences in the experience during pregnancy and delivery between mothers and fathers relate to bonding. Have they been studied in the context of bonding in previous literature? Are those variables related to bonding in mothers? I’m having trouble seeing how this is evidence of the likelihood of bonding problems with the child.

-The authors reference one other review on paternal bonding – the list the limitations of that study, but don’t actually provide any information on what the results of the past study were. It would be useful to describe the results of that review to show how the current study can extend those results.

-As a reader, I would find it helpful if the authors could further describe “bonding problems.” What might lead to such bonding problems? Are there common elements to bonding problems, such as parents who are separated from their newborns due to NICU or other reasons?

-It would also be useful for the authors to further explain the one-sided nature of bonding. The authors state that bonding in this context is really only about the parent’s bonding, not the reciprocal bond between parent and child. Is it still considered successful/healthy bonding if the parent is bonded but the child is not?

-Reliability of past assessment measures is listed as a problem, but no information about how paternal bonding has been assessed in the past is given. It would be useful to provide this information.

Method:

-In general, it would be helpful if the authors could clarify exactly what they mean by “the constructs of  the concept of paternal bonding.” In addition, the second two bullet points in research question 1 could be clarified, as they are quite broad without any specific details: “What are consistent in the use and meaning?” and “Is there clarity regarding the differentiation of concepts from other related concepts?” For instance, the authors state in the discussion that question about differentiating concepts was refer to differentiating paternal and maternal bonding – this could be added into the question to make it more specific.

-with regard to the search for articles, can the authors provide context for why they only manually searched two of the databases in 2021?

Results:

-The results section is well-organized

-When the authors state that “The most common definition of bonding was a description regarding affective or emotional ties” (line 200), it would be useful to provide some examples of affective or emotional bonding. Similarly, providing examples of the cognitive and behavioral elements would make this part clearer.

-more details about how perception of the parent’s child-rearing history, neglect, and trauma could influence paternal bonding would be useful (lines 247-251). What specifically about those things seemed to influence bonding and in what way?

-Table 2 might be better suited for the supplemental materials. It’s a lot of information for the reader. It might be more effective for the authors to instead provide a more detailed description of what they took from that table, rather than provide every detail in the table.

Discussion

-The discussion was detailed and comprehensive, with useful conclusions.

-In table 1, the authors noted important details like the study location, design, and form of data collection. It seems like some discussion of how these variables could influence the results is warranted. For instance, does the definition of bonding differ by country? Does an experiment tap into different constructs than a qualitative interview? Etc.  

Author Response

Thank you for your important comments regarding our manuscript titled, “The current concept of paternal bonding: A systematic scoping review,” and for allowing us to submit a revised draft. We appreciate and have carefully considered all the reviewers’ comments and provided answers to each of them. Please find all the comments and answers below; modified parts as shown with yellow highlights in the manuscript.

We are very grateful for the review and appraisal of our scoping review and how it contributes to this area. Each of your comments encouraged us, and also improved the quality of the manuscript.

We appreciate your contribution and effort.

According to all of your comments, we modified the manuscript as follows:

Introduction:

  • -I found the introduction somewhat hard to follow and many of the claims did not feel well-supported. For instance, the authors state “Thus, bonding, a special emotional tie, has different concepts and constructs for fathers and mothers” (line 96) but shortly thereafter state “Additionally, the difference between paternal and maternal bonding remains unclear” (line 106). More evidence needs to be provided to explain how and why bonding differs between mothers and fathers or, if it’s the case that we actually don’t know that, the claim that there are differences should not be made.
    • We rephrased lines 96 to 100 below;
      • Based on these findings, fathers may have experiences with their children that differ from those of mothers. Hence, they may have unique emotions toward their children. Thus, we hypothesized that the concepts and constructs of bonding vary between fathers and mothers. However, unfortunately, there is limited information to determine our hypothesis.

  • -Related, I was unclear how the differences in the experience during pregnancy and delivery between mothers and fathers relate to bonding. Have they been studied in the context of bonding in previous literature? Are those variables related to bonding in mothers? I’m having trouble seeing how this is evidence of the likelihood of bonding problems with the child.
    • We added some explanations in lines 72 to 78 shown below.
      • Due to the significance of the bonding process in parenting, the focus of perinatal mental health research has shifted from perinatal depression to perinatal bonding and bonding disorders [13, 14]. Most prior studies have focused on maternal bonding, with limited studies on paternal bonding. Although the child’s main caregivers are both the father and mother, the parenting involvement differs between men and women. Moreover, there are biological role differences at play, such as in childbirth and direct breastfeeding.
    • Also, from your comments above that are related to our RQ, it might lead to confusion if we described them in detail. Therefore, we considered the description in the introduction section.

  • -The authors reference one other review on paternal bonding – the list the limitations of that study, but don’t actually provide any information on what the results of the past study were. It would be useful to describe the results of that review to show how the current study can extend those results.
    • We added P3, lines 106 to 110, as shown below:

Furthermore, there is only one integrative review regarding the factors and interventions influencing early father–infant bonding [18]. This review revealed the fathers’ adjustment and transition, influencing variables of father–infant bonding, and promotional interventions to encourage father–infant bonding. Additionally, it described the importance of support, especially during childbirth and transitioning to being a father, associated with the father–infant bonding. However, each study included in this review did not consistently define paternal bonding.

  • -As a reader, I would find it helpful if the authors could further describe “bonding problems.” What might lead to such bonding problems? Are there common elements to bonding problems, such as parents who are separated from their newborns due to NICU or other reasons?
    • We added P3, lines 68 to 69, as shown below:
      • In previous studies, parents with bonding problems, such as impaired bonding, showed poor interaction with their children [8, 9].

  • -It would also be useful for the authors to further explain the one-sided nature of bonding. The authors state that bonding in this context is really only about the parent’s bonding, not the reciprocal bond between parent and child. Is it still considered successful/healthy bonding if the parent is bonded but the child is not?
    • We recognized that the current definition refers to a tie from the parent, and not a mutual one with the child. Thus, we followed the definition that bonding is the tie from the parents toward the child, and refer to studies that described it as such.

  • -Reliability of past assessment measures is listed as a problem, but no information about how paternal bonding has been assessed in the past is given. It would be useful to provide this information.
    • This part is the RQ of this study. However, the consistency of evaluations using a unified measurement scale has yet to be demonstrated in previous papers, and its description was unclear.

Thus, we added some explanations in P3, lines 115 to 117, as shown below:

  • For example, the scales for maternal bonding should not be used for fathers if the concepts of maternal and paternal bonding are different.

Method:

  • -In general, it would be helpful if the authors could clarify exactly what they mean by “the constructs of  the concept of paternal bonding.” In addition, the second two bullet points in research question 1 could be clarified, as they are quite broad without any specific details: “What are consistent in the use and meaning?” and “Is there clarity regarding the differentiation of concepts from other related concepts?” For instance, the authors state in the discussion that question about differentiating concepts was refer to differentiating paternal and maternal bonding – this could be added into the question to make it more specific.
    • We added further explanations in lines 134 to 135 and 136 to 137 on P3, as shown below:
      • What terms are consistent with the use and meaning of paternal bonding? Do they have the same use and meaning as maternal bonding?
      • Is there clarity regarding the differences between the concept and other related concepts such as maternal bonding or attachment?
      • Are these the same as maternal bonding, or are there unique elements or functions in paternal bonding? (RQ2; lines 139 to 140)

  • -with regard to the search for articles, can the authors provide context for why they only manually searched two of the databases in 2021?
    • We found a paper later, which was not included in the first search performed in 2020. Therefore, we hand-searched databases to determine studies based on the reference lists of the included studies.

Therefore, we modified it as below.

  • The manual search was conducted on PubMed and Google Scholar by checking the reference lists of included studies in January 2021.

Results:

  • -When the authors state that “The most common definition of bonding was a description regarding affective or emotional ties” (line 200), it would be useful to provide some examples of affective or emotional bonding. Similarly, providing examples of the cognitive and behavioral elements would make this part clearer.
    • We agree with your comment. They are all shown in Table 2. There seems to be considerably information, but the details would be helpful to the readers, so we added them below in lines 203 to 204.
      • All the descriptions are shown in Table 2.

  • -more details about how perception of the parent’s child-rearing history, neglect, and trauma could influence paternal bonding would be useful (lines 247-251). What specifically about those things seemed to influence bonding and in what way?
    • This is the result of the study we reviewed. Unfortunately, we did not find detailed descriptions in the study. Thus, we could not explain this specifically; however, we added "to negative paternal bonding" on line 261 instead.

  • -Table 2 might be better suited for the supplemental materials. It’s a lot of information for the reader. It might be more effective for the authors to instead provide a more detailed description of what they took from that table, rather than provide every detail in the table.
    • We carefully considered your comment. However, we are concerned that if we described this further in the manuscript instead of only in Table 2, it may result in excessive information for the readers. Additionally, as I mentioned above, Table 2 is helpful for readers. Thus, we decided to retain Table 2, and not provide it as a supplementary file.

Discussion

  • The discussion was detailed and comprehensive, with useful conclusions.
    • We appreciate your comments, which have encouraged us.

  • -In table 1, the authors noted important details like the study location, design, and form of data collection. It seems like some discussion of how these variables could influence the results is warranted. For instance, does the definition of bonding differ by country? Does an experiment tap into different constructs than a qualitative interview? Etc.  
    • We discussed the findings according to Table 1 in the discussion section (P22, lines 564 to 572 and lines 597 to 599), as shown below:
      • First, as shown in Table 1, the study locations were spread worldwide. This implies that paternal bonding is of global interest, but most studies were conducted with a similar cultural background. Although the study locations were quite diverse, previous studies did not explicitly indicate the cultural aspect of paternal bonding. To our knowledge, no global studies published elsewhere regard paternal bonding across cultures. By and large, studies cited common definitions not considering the cultural context. Nevertheless, we acknowledge the critical role of culture in paternal bonding. Besides, various measurement tools were used across studies and cultures. Hence, the measurement invariance of the scale is essential, and further research exploring these areas is needed.
      • Some qualitative studies provided fathers' direct responses and that method might lead to identifying the common concept of paternal bonding.

Reviewer 2 Report

Dear authors,

Thank you for letting me review this very interesting paperThe current concept of paternal bonding: A systematic scoping review”.

I miss in title bonding with their infant, it is not obvious to whom the bonding is.

According to whom and how? Page 2 line 75 Although the child´s main caregivers are both the father and the mother, parenting involvement between men and women differ.

 Please be consistent if you have parental or maternal first or second.

Please reduce your use of show

I wonder why you have not included other used instruments in the studies. For instance you write on page 22 line 556 there are no reports regarding father´s bonding disorders as mental health problem. However, have at least one study (Kerstis, 2016) included the correlation between bonding and depressive symptoms.

Author Response

              Thank you for your important comments regarding our manuscript titled, “The current concept of paternal bonding: A systematic scoping review,” and for allowing us to submit a revised draft. We appreciate and have carefully considered all the reviewers’ comments and provided answers to each of them. Please find all the comments and answers below; modified parts as shown with yellow highlights in the manuscript.

Thank you for your comments and efforts in reviewing our manuscript.

We carefully considered all comments and have modified the manuscript as follows:

  • I miss in title bonding with their infant, it is not obvious to whom the bonding is.
    • According to our review, paternal bonding is not only relevant for infants. Some studies showed that it begins during pregnancy and includes feelings towrd a fetus as well. Thus, we have not described who the concept of paternal bonding is directed at.

  • According to whom and how? Page 2 line 75 Although the child´s main caregivers are both the father and the mother, parenting involvement between men and women differ.
    • Thank you for your comments. This describes the parents' care for their children, and the difference between mothers and fathers is that mothers provide care more directly, such as breastfeeding, while fathers do not. However, this was not mentioned clearly and we modified the text as follows:
      • Due to the significance of the bonding process in parenting, the focus of perinatal mental health research has shifted from perinatal depression to perinatal bonding and bonding disorders [13, 14]. Most prior studies have focused on maternal bonding, with limited studies on paternal bonding. Although the child’s main caregivers are both the father and mother, the parenting involvement differs between men and women.

  • Please be consistent if you have parental or maternal first or second.
    • We fixed the text to show maternal first consistently throughout the manuscript..

  • Please reduce your use of show
    • We have tried to reduce the usage of “show” in the manuscript. (Highlights in P2 L70, P18 L364, P 19 L376, and P19 L404

  • I wonder why you have not included other used instruments in the studies. For instance you write on page 22 line 556 there are no reports regarding father´s bonding disorders as mental health problem. However, have at least one study (Kerstis, 2016) included the correlation between bonding and depressive symptoms.
    • This review aimed to clarify the concept of paternal bonding, and not other related mental health concepts. Moreover, bonding disorder might be considered a disease, depending on other mental health problems. Therefore, we did not include other instruments in our paper.
